# Larger Medial Contact Area and More Anterior Contact Position in Medial-Pivot than Posterior-Stabilized Total Knee Arthroplasty during In-Vivo Lunge Activity

**DOI:** 10.3390/bioengineering10030290

**Published:** 2023-02-23

**Authors:** Diyang Zou, Jiaqi Tan, Nan Zheng, Zhi Ling, Wanxin Yu, Ming Han Lincoln Liow, Yunsu Chen, Tsung-Yuan Tsai

**Affiliations:** 1School of Biomedical Engineering & Med-X Research Institute, Shanghai Jiao Tong University, Shanghai 200030, China; 2Engineering Research Center of Digital Medicine and Clinical Translation, Ministry of Education, Shanghai 200030, China; 3Shanghai Key Laboratory of Orthopaedic Implants & Clinical Translation R&D Center of 3D Printing Technology, Department of Orthopaedic Surgery, Shanghai Ninth People’s Hospital, Shanghai Jiao Tong University School of Medicine, Shanghai 200010, China; 4Department of Orthopedics, Shanghai Sixth People’s Hospital, Shanghai Jiao Tong University, Shanghai 200233, China; 5Department of Orthopaedic Surgery, Singapore General Hospital, Singapore 169608, Singapore; 6Stavros Niarchos Foundation Complex Joint Reconstruction Center, Hospital for Special Surgery, New York, NY 10021, USA; 7TAOiMAGE Medical Technologies Corporation, Shanghai 200120, China

**Keywords:** total knee arthroplasty, medial pivot, posterior stabilized, lunge, in-vivo kinematic, articular contact

## Abstract

This study aimed to compare the in-vivo kinematics and articular contact status between medial-pivot total knee arthroplasty (MP-TKA) and posterior stabilized (PS) TKA during weight-bearing single-leg lunge. 16 MP-TKA and 12 PS-TKA patients performed bilateral single-leg lunges under dual fluoroscopy surveillance to determine the in-vivo six degrees-of-freedom knee kinematics. The closest point between the surface models of the femoral condyle and the polyethylene insert was used to determine the contact position and area. The nonparametric statistics analysis was performed to test the symmetry of the kinematics between MP-TKA and PS-TKA. PS-TKA demonstrated a significantly greater range of AP translation than MP-TKA during high flexion (*p* = 0.0002). Both groups showed a significantly greater range of lateral compartment posterior translation with medial pivot rotation. The contact points of PS-TKA were located significantly more posterior than MP-TKA in both medial (10°–100°) and lateral (5°–40°, 55°–100°) compartments (*p* < 0.0500). MP-TKA had a significantly larger contact area in the medial compartment than in the lateral compartment. In contrast, no significant differences were observed in PS-TKA. The present study revealed no significant differences in clinical outcomes between the MP and PS groups. The PS-TKA demonstrated significantly more posterior translations than MP-TKA at high flexion. The contact points are located more posteriorly in PS-TKA compared with MP-TKA. A larger contact area and medial pivot pattern during high flexion in MP-TKA indicated that MP-TKA provides enhanced medial pivot rotation.

## 1. Introduction

Total knee arthroplasty (TKA) is an effective surgical procedure for end-stage knee osteoarthritis in terms of functional restoration of the knee and pain relief [1,2,3,4]. Despite this, over 20% of patients complained about continuous pain and dissatisfactory function [4]. One of the reasons is that the replaced articular surface changes the kinematics of the knee [5,6,7].

The posterior stabilized (PS) TKA was reported to be one of the most commonly used implant types, which relies on the cam-post mechanism to maintain anterior-posterior stability and increase the femoral rollback [8,9,10]. Previous studies have investigated that the PS-TKA demonstrated paradoxical anterior femoral translation with increasing knee flexion and lateral pivot movement pattern during gait [11,12]. The medial-pivot (MP) TKA is a relatively new design that aims to restore the normal knee kinematic [13,14]. The ball-and-socket medial compartment provides knee stability, and the relatively flat surface on the lateral compartment allows for greater anterior-posterior translation. Multiple studies have reported superior clinical patient outcomes in MP-TKA compared to PS-TKA [1,15,16,17,18]. In contrast, some other studies have shown no difference between the two groups [2,19,20,21]. Also, the comparison of in-vivo six-degree-of-freedom (6-DOF) knee kinematic during daily activity such as gait [3,14,22], stair climbing [23], sit-to-stand [24] between MP-TKA and PS-TKA have shown inconsistent results. However, these activities only study a limited range of knee flexion, and more challenging weight-bearing, deep-flexion type activities such as lunge should be quantified to assess the difference between MP-TKA and PS TKA.

Articular contact kinematics, which includes the movement pattern of the medial/lateral contact point or pivot point location, is highly associated with patient satisfaction and predicts polyethylene wear [3,9,25]. Numerous studies used cadaveric specimens or finite element methods to compare the contact mechanics among different implants under controlled conditions [26,27,28]. In-vivo articular contact forces have also been measured using embedded force sensors [29,30]. Dynamic fluoroscopic imaging technique has been widely used to detect in-vivo femoral condyle motion in healthy subjects and TKA patients [7,24,31]. Previous studies have been limited to analyzing changes in 6-DOF kinematics and projected contact points on the polyethylene articulation [14,24]. However, no clinical study has compared contemporary MP-TKA with PS-TKA in terms of in-vivo articular contact or pivot center location on tibial polyethylene. Thus, the difference in articular contact kinematics of MP-TKA and PS-TKA is unclear. Related to this, polyethylene wear prediction in these two designs requires further investigation.

This study aimed to investigate the in-vivo kinematics, articular contact positions, pivot point location and contact area of the medial and lateral compartments in MP-TKA and PS-TKA patients during weight-bearing single-leg lunge activity using a dual fluoroscopic imaging tracking system (DFIS).

## 2. Materials and Methods

### 2.1. Patient Demographics

This study was carried out after the approval of the Ethics Committee of Shanghai Sixth People’s Hospital, China (Protocol Code: YS-2018-124). In this study, 16 patients who had received the medial-pivot cruciate-substituting (MP CS, Evolution, MicroPort Orthopedics, Arlington, TN, USA) total knee arthroplasty and 12 patients who had received posterior-stabilized (Genesis II, Smith and Nephew, Memphis, USA) total knee arthroplasty were recruited. A senior doctor randomly determined the implant type for those patients with similar clinical indications. Consistent inclusion and exclusion criteria were applied in the two groups. The inclusion criteria were as follows: (1) 18–85 years; (2) diagnosed with end-stage osteoarthritis without neuromuscular disease (Kellgren–Lawrence grade 3 and grade 4); (3) agreeing to participate in this research and signing an informed consent form before the experiment. The exclusion criteria: (1) diagnosed with valgus deformation or varus deformation over 10 degrees before TKA surgery; (2) any postoperative complications, such as unbearable pain or instability, etc., before the experiment. (3) body mass index > 40 kg/m^2^, (4) rheumatoid arthritis. (5) chronic inflammatory joint diseases, (6) patients with a pre-pathological abnormal gait. (7) pregnancy or breastfeeding. One senior surgeon performed all TKAs using a standard surgical technique which included a midline incision, parapatellar exposure, distal femoral resection, posterior cruciate ligament sacrifice, gap balance and tibial resection to achieve mechanical alignment [32,33]. The patient’s demographic data are shown in Table 1.

### 2.2. Clinical Outcome Evaluation

Clinical outcome evaluation was performed using the following questionnaires: (1) The Knee Society Score system (KSS) included “Knee Score” and “Function Score” to assess the knee clinical symptoms and individual’s functionality [34]. (2) The EQ-5D is a five-item questionnaire to assess the non-knee-specific quality of life with three levels. (3) The patient satisfaction level about the effect of TKA surgery with a 4-level rating (very dissatisfied, dissatisfied, satisfied and very satisfied, scored with 1–4 points) [2]. (4) The Forgotten Joint Score (FJS) assesses the feeling of the existence of a prosthesis when carrying out daily activities. A higher final score indicates better clinical performance.

### 2.3. 3D Reconstruction of Knee Models

All patients underwent a full-length computer tomography (CT) scan (Discovery CT750 HD, GE Medical System, Chicago, IL, USA, 120 kVp) from the hip to the ankles postoperatively with an image resolution of 512 × 512 pixels and voxel size of 0.86 × 0.86 × 0.63 mm^3^. Images were imported into medical image software (Amira 6.7.0, Thermo Fisher Scientific, Waltham, MA, USA) to reconstruct the three-dimensional (3D) surface models of bone (femur and tibia) and TKA implant (femoral component and tibial component). Anatomical coordinate systems of the non-operated contralateral knee were built according to bone landmarks first [35]. The anatomical coordinate systems of the operated side were mirrored from the non-operated side through previously published and validated surface-to-surface registration methods (Figure 1A) [7,14]. The coordinate systems of high-precision computer-aided design (CAD) TKA components were built according to the original design of the manufacturer (Figure 1B,C). The high-precision CAD models of the MP-TKA and PS-TKA prostheses obtained from the manufacturers or laser scanning were used to align with CT reconstructed model [7,36]. The relative positions of the implant and femur or tibia were then determined for kinematic analysis.

### 2.4. In-Vivo Kinematic Measurements

All MP-TKA and PS-TKA patients performed weight-bearing single leg lunge from full extension to maximal flexion on the operated side under a dual fluoroscopic imaging system (BV Pulsera, Philips, Andover, MA, image resolution 1024 × 1024 pixels, dynamic image frequency 30 Hz). The approximate time of a full range of lunge was about 2 s with 60 images. The 2-dimensional dynamic fluoroscopic images obtained from 2 views and 3D precise CAD models of TKA components were imported into our customized program (MATLAB, MathWorks, Natick, MA, USA) to determine the in-vivo 3D relative position between the femoral and tibial component. By matching the metal femoral and tibial component contour with both dual dynamic fluoroscopic images at each frame in a reconstructed virtual environment, the position of the femoral and tibial components in 3D space was determined. The polyethylene insert position was determined from the position of the tibial component according to their fix-bearing lock mechanism. The accuracy of the DFIS tracking technique was previously validated with 0.11 mm for translation and 0.24° for rotation [35]. The flexion/extension, varus/valgus and internal/external (I/E) rotation of the tibia with respect to the femur were calculated according to the Euler angle transformation with a “z-x-y” order (Figure 1A). The anterior/posterior (AP: *x*-axis), proximal/distal (PD: *y*-axis) and medial/lateral (ML: *z*-axis) of the femur with respect to the tibia were calculated (Figure 1A). In-vivo kinematics of the TKA was then quantified during the whole flexion path.

### 2.5. In-Vivo Tibiofemoral Articular Contact Measurement

The local TKA component coordinate systems were used to illustrate the change of tibiofemoral contact along the flexion path (Figure 1B,C). The penetration method was used to define the contact status by calculating the closest normal distance from each polyethylene insert mesh to the condyle mesh [37]. The area where the face-to-face distance of less than 1 mm was considered to be defined as the contact area and the mean of all contact facet centers was used to determine the contact point [38]. The random Gaussian noises of DFIS matching error were added to the femoral and tibial components to evaluate the sensitivity of contact measurement [35]. The average contact location error was 0.2 ± 0.4 mm in 3D space. The regions of contact in medial and lateral compartments were calculated with a 5° flexion angle increase interval. The overlapping region of each contact area was counted to generate the heatmap of cumulative contact times. The percentage of contact region with respect to medial and lateral polyethylene surface was measured (Figure 1B,C). Specifically, the data were divided into 4 ranges: low flexion (0°–30°), pre-middle flexion (30°–60°), post-middle flexion (60°–90°) and high flexion (90°–100°) for lunge kinematic analysis. The location of the pivoting point was determined as the average intersect point of consecutive lines linking the medial and lateral contact points at different ranges of lunge flexion angle by using the least-squares optimization method [39]. All the contact trajectories and contact regions were normalized and then mapped onto a polyethylene insert with a standard type of 4R (MP: length 70.6 mm, width 47.3 mm, medial area, 1014.8 mm^2^ and lateral area 1149.7 mm^2^. PS length 68.5 mm, width 44.4 mm, medial area, 916.9 mm^2^ and lateral area 915.6 mm^2^).

### 2.6. Statistical Analysis

A post-hoc power analysis was performed to estimate the statistical power between MP-TKA and PS-TKA groups (G*Power version 3.1; Franz Faul, Universität Kiel, Germany). Average values and standard deviations were used to describe all variables. The Wilcox rank sum test was used to compare the difference in clinical outcome, in-vivo kinematics and contact point position between MP TKAs and PS TKAs. A Mann-Whitney U test was used to compare the difference between medial and lateral contact areas in MP-TKA and PS-TKA. All the statistical analyses were performed using MATLAB (MATLAB 2020a, MathWorks, Natick, MA, USA). The significant difference level was set at *p* < 0.05.

## 3. Results

### 3.1. Clinical Outcomes

The preoperative and postoperative hip-knee-angle of MP-TKA and PS-TKA were measured using standard X-ray images, showing no significant differences. There were no significant differences in clinical scores between the MP and PS groups (Table 2).

### 3.2. In-Vivo Kinematic of TKA

Both groups had paradoxical roll forward during the initiation of a lunge with the peak value of −3.1 ± 4.9 mm at 22° of flexion in MP-TKA and that of −2.7 ± 3.6 mm at 24° of flexion in PS-TKA. Then, both groups moved posteriorly to the end of flexion (MP: −7.8 ± 3.1 mm, PS: −12.4 ± 2.9 mm, *p* = 0.0008, effect power: 0.99). The average ranges of AP translation in the MP and PS groups were 5.6 ± 2.5 mm and 10.4 ± 2.9 mm during the whole flexion (*p* = 0.0002, effect power: 0.99). Both groups initially demonstrated external tibial rotation till around 27°–28° of flexion (MP: 0.4° ± 4.8°, PS: −0.6° ± 5.6°, *p* = 0.6313). Subsequently, both groups showed internal tibial rotation until the end of flexion (MP: 8.4° ± 5.2°, PS: 3.8° ± 6.4°, *p* = 0.1101). The average and range of I/E rotation in MP and PS groups were 9.7° ± 3.6° and 8.3° ± 2.8° with no significant differences (*p* = 0.3238) (Figure 2A,B).

### 3.3. In-Vivo Contact Position of TKA

In the MP-TKA groups, the contact point moved posteriorly until the 20° of flexion and then moved anteriorly until the 100° of maximal flexion with an AP translation range of 6.8 ± 3.1 mm (13.8 ± 6.4%) in the medial compartment (Figure 3B, Figure 4B and Table 3). The contact position of the lateral compartment continuously moved posteriorly during lunge with an AP translation range of 9.3 ± 4.4 mm (19.0 ± 9.1%; Figure 3A, Figure 4A and Table 3). The range of AP translation on the lateral side was significantly larger than the medial side in MP-TKA (*p* = 0.0261, effect power: 0.53). In the PS-TKA group, the contact point moved posteriorly until the 25° of flexion, then moved anteriorly, and then kept relatively stable at 35°–70° of flexion, and finally moved posteriorly until the maximal flexion of 100° with AP translation range of 6.6 ± 2.3 mm (14.9 ± 5.2%) in the medial compartment (Figure 3B, Figure 4B and Table 3). In the lateral compartment, the contact point continuously moved posteriorly with an AP translation range of 10.5 ± 4.1 mm (23.7 ± 9.3%; Figure 4A,B and Table 3), which is significantly larger than the medial side (*p* = 0.0009, effect power: 0.83).

The contact point’s location on PS-TKA showed significantly more posterior than on MP-TKA in both medial (10°–100°) and lateral (5°–40° and 55°–100°) compartments (Figure 4, *p* < 0.0500). Significant differences in AP translation range were observed between MP-TKA and PS-TKA in both medial (MP: 6.5 ± 5.0%, PS: 12.3 ± 4.4%, *p* = 0.0026, effect power: 0.89) and lateral (MP: 8.7 ± 5.2%, PS: 13.7 ± 5.1%, *p* = 0.0373, effect power: 0.69) compartment from post-middle flexion to high flexion (60°–100°). The average pivoting point of PS-TKA was located in the medial compartment and close to the medial edge during the whole flexion (Posterior: −12.7 ± 4.2%, Medial: −44.8 ± 29.3%), while it was in the medial compartment and close to the center in MP-TKA (Posterior: −3.6 ± 1.7%, Medial: −4.3 ± 15.9%). The medial-pivoting was observed during the post-middle flexion in MP-TKA and the low flexion, the post-middle flexion and the high flexion in PS-TKA (Figure 3 and Table 3).

### 3.4. In-Vivo Contact Area of TKA

The average merged areas where contact had occurred on the polyethylene insert at any flexion interval were 48.7 ± 6.4% in the medial and 56.7 ± 11.2% in the lateral compartment for PS-TKA (Figure 5A and Table 4). No significant difference was observed for the summed area between the medial and lateral compartments of PS-TKA (*p* = 0.0639). In contrast, the contact area showed a significant difference between the lateral compartment (46.1 ± 7.9%) and medial compartment (71.5 ± 5.3%) in MP-TKA (*p* = 0.0197 and effect power: 0.99) (Figure 5B and Table 4). The areas where contact at all flexion intervals (21 times) were 10.4 ± 5.1% in the medial and 3.4 ± 2.0% in the lateral compartment on PS-TKA and were 23.1 ± 8.8% in the medial and 12.6 ± 4.3% in the lateral compartment on MP-TKA (Figure 5 and Table 4). Significant differences were observed between the medial and lateral areas with 21 times contact in both MP-TKA (*p* < 0.0001, effect power: 0.99) and PS-TKA (*p* = 0.0136, effect power: 0.99).

## 4. Discussion

The most important finding of the present study was that a larger contact area was observed in the medial compartment of MP-TKA compared with PS-TKA during flexion. The PS-TKA demonstrated parallel posterior translation, whereas MP-TKA had pivot movement with medial contact position keeping relatively stable and lateral contact position moving posteriorly from post-middle to high flexion. There were no significant differences in tibial axial rotation motion between groups, with both groups demonstrating initial external rotation followed by an internal rotation.

Our finding about contact area indicated that MP inserts help to keep the implants rotating on the medial side compared to PS implants (Figure 5 and Table 4). Also, the articular contact area was regarded to reflect contact stress indirectly [38,40]. Murakami et al. reported a similar contact area position of the Genesis Ⅱ PS-TKA through an in-vitro knee simulation experiment [27]. Steinbruck et al. reported a significantly higher contact area and lower peak pressure of MP-TKA compared with PS-TKA in the medial compartment, which was consistent with our result [28]. Shu et al. also reported a similar contact area and lower peak contact pressure in MP design TKA through the FEA method [26]. The incongruous loading conditions and soft tissue restraints may cause inconsistent results between previous in-vitro experiments and our in-vivo results [27,28]. Minoda et al. analyzed the synovial fluid of the knee from TKA patients and found that the total number of small polyethylene particles in the MP design was less than half of that in the PS design. This finding suggests that the MP tibial insert had superior in-vivo wear resistance than the PS design [41]. However, Wimmer et al. suggested more conformity may increase the risk of surface fatigue damage due to more possibility of edge loading and entrapped bone cement [42]. Therefore, these results suggest that having more contact on the medial component probably means less pistoning and a smoother arc of motion throughout the flexion of the knee. Further long-term follow-up and retrieval analysis studies are required to provide more evidence.

In the current study, the contact trajectories of the lateral component showed a similar consecutive posteriorly translated pattern during the whole flexion between MP-TKA and PS-TKA. Also, a larger range of AP translation in the lateral component than that in the medial was found in both groups. But it is still smaller than the normal subject, whether on the medial or lateral side [6,43,44]. These findings indicated that the medial-pivot design and post-cam design both partially reproduced the normal lateral articular contact pattern. Furthermore, our study found that the contact positions of PS-TKA were located in the posterior of the polyethylene insert and showed more posterior than MP-TKA at most ranges of flexion (Figure 4). The posterior contact may result in an increased shear force on the surface of the polyethylene insert and lead to delamination damage [45]. Retrieved PS polyethylene inserts showed greater volumetric loss on the posterior part of the insert [46,47], which is consistent with our contact kinematic result of PS-TKA. Although the PS-TKA showed greater posterior translation than MP-TKA during high flexion, it also increased the stress and wearing of the post-cam structure, which was proved to be the main factor leading to the break of the cam [48]. Our results suggest that the raised lip at the post of lateral tibial insertion in MP design limited the further posterior translation during high flexion (Figure 3B). Still, it might provide more knee stability.

One of the highlights of this study was to evaluate the pivot pattern of the MP-TKA during lunge and compare it with traditional PS-TKA. The pivoting point of the whole flexion range was closer to the center of the polyethylene plateau rather than we would expect at the lowest point of the ball-and-socket structure in MP-TKA [49]. In addition, the lateral pivot pattern appeared in the partial phase of flexion in our study. Medial pivoting has been reported by Gray et al., who studied MS-TKA (medial-stabilized GMK sphere, Medacta International, Castel San Pietro, Switzerland) during gait [3]. Nevertheless, Miura et al. have reported the coexistence of medial and lateral pivot patterns of MP-TKA during the early stance phase of gait [50]. With respect to the pivot pattern of PS-TKA during the lunge, our study showed a similar medial pivot to previous studies [12,51]. But other studies also revealed the lateral pivot pattern of PS-TKA [3,12,25]. Remarkably, both medial and lateral pivot patterns occur among healthy subjects [39,44,52,53]. The pivot point is mainly determined by the kinematics of the knee, which are highly associated with the type of movement [54], the implant type [3,22] and the individual characters of subjects [50].

Previous fluoroscopic image studies observed the 6-DOF differences between MP-TKA and PS-TKA during daily activities. Schutz et al. reported that the similar MP design exhibited a significantly larger range of internal/external rotation when compared with the PS design during level walking (MP: 11.9°, PS: 10.5°), downhill walking (MP: 11.5° PS: 8.9°) and stair descent (MP: 13.2° PS: 9.0°) [22]. However, Tan J et al. demonstrated a similar kinematic pattern between MP and PS during gait [14]. Specifically, our study first showed larger posterior translation in PS-TAK than in MP-TKA at a high flexion range. Although greater femoral rollback has been suggested to improve patient satisfaction [17], no differences in clinal outcome were found in our results (Table 2). The abnormal anterior translations were observed at early flexion in both groups (Figure 2). Roberti et al. also reported abnormal femoral anterior translation, and it was greater in the PS-TKA compared with the MP-TKA [24]. This situation has been reported in-vivo kinematic studies of TKA with anterior cruciate ligament (ACL) resection [55,56,57] and in ACL deficient patients [58]. The continuous femoral rollback was observed in the normal knee [6,43], bi-cruciate retained TKAs [31,59] and fixed-bearing unicompartmental knee arthroplasty patient [60] during flexion. Those indicated that the post-cam design or medial pivot design can only partially restore the normal knee kinematic and cannot fully replace the function of ACL in anterior stability at the early flexion phase.

It is not easy to draw a conclusion that one implant design is superior to another from the current study. Rao et al. reported that several movement characteristics are highly associated with bad TKA outcomes and unsatisfactory, such as abnormal anterior translation (larger than −8.5% ± 4.4%) on the medial side of the knee during early flexion of lunge (0°–30°), less posterior translation (3.5% ± 4.5%) on the lateral side in deep flexion (60°–90°) [5]. Both of our contact kinematic results in two types of implant are higher than the reported bound value and with good clinical outcomes. Meneghini et al. claimed that the traditional PS design with post-cam is no longer necessary for primary TKA [61]. Our result indicated that the medial-pivot design, which does not rely on the cam–post mechanism, could be the alternative for subjects and provide high medial stability. But the kinematic difference and contact difference at high flexion of lunge should be noticed.

Some potential limitations should be mentioned in our study. First, only lunge activity was investigated to compare the difference between MP-TKA and PS-TKA. However, the kinematic differences in some daily activities, such as level walking or sit-to-stand, have been evaluated in previous studies [3,14,20,24]. Lunge activity was considered to evaluate the implant performance better, especially under high flexion. Second, the in-vivo contact stress distribution on polyethylene cannot be evaluated in this study. The contact trajectories or areas are one of the methods to predict the in-vivo articular surface wear. Clinically relevant polyethylene wear and damage will require further investigation through long-term follow-up retrieval studies. Third, the influence on the post-cam mechanism of PS-TKA was inferred from the contact position on the tibial polyethylene. Further study should be investigated to evaluate the in-vivo biomechanical behavior of the post-cam of PS-TKA.

## 5. Conclusions

In conclusion, our study quantified the in-vivo kinematic, articular contact trajectory and contact area of MP-TKA and compared those with traditional PS-TKA during the lunge activity. There were no significant differences in clinical outcomes between the MP-TKA and PS-TKA groups. PS-TKA demonstrated significantly more posterior translation than MP-TKA at high flexion. The articular contact points are located more posteriorly in PS-TKA compared with MP-TKA. A larger contact area and medial pivot pattern during high flexion in MP-TKA indicated that MP-TKA provides enhanced medial pivot rotation.

## Figures and Tables

**Figure 1 bioengineering-10-00290-f001:**
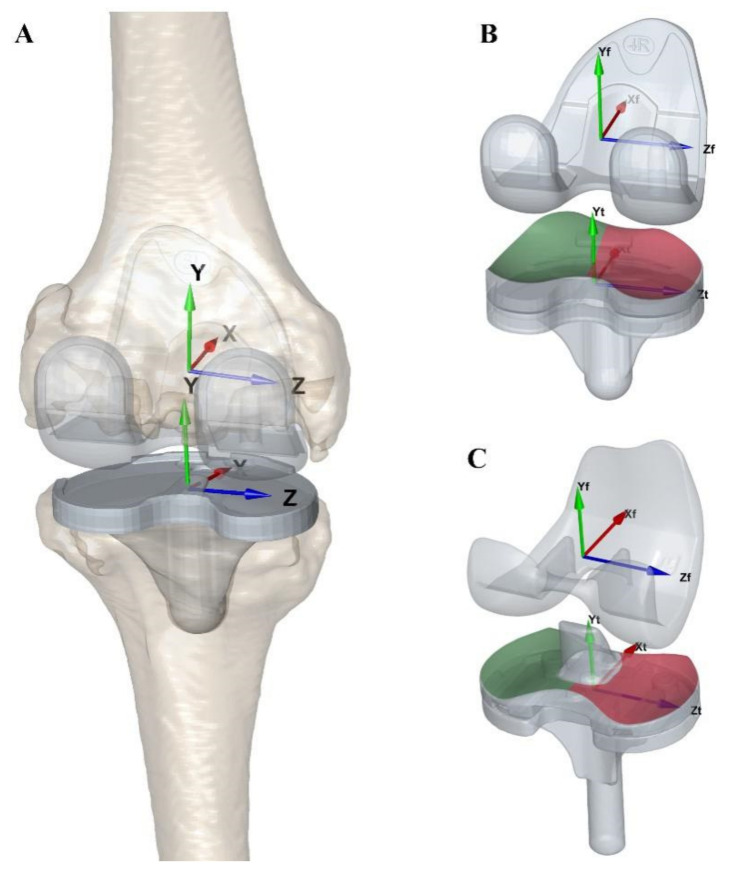
(**A**) The Anatomical coordinate system definition of right TKA patients. The origin of the femur was defined as the midpoint of the clinical transepicondylar axis (c-TEA). The origin of the tibia was defined as the center of the tibial plateau. (**B**) The local coordinate systems of MP-TKA components. (**C**) The local coordinate systems of PS-TKA components. The red (X-axis), green (Y-axis) and blue (Z-axis) arrows indicated the anterior/posterior (AP), proximal/distal (PD) and medial/lateral (ML) axes, respectively. The tibial polyethylene (PE) insert was divided into medial up-surface (green area) and lateral up-surface (red area) according to the local sagittal plane. MP: Medial-pivot. PS: Posterior-stabilized. TKA: Total knee arthroplasty.

**Figure 2 bioengineering-10-00290-f002:**
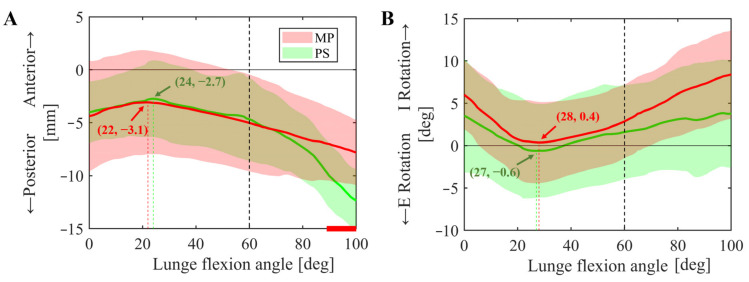
In-vivo kinematics of the MP-TKA and the PS-TKAs during the single-leg lunge. (**A**) The anterior and posterior translations change of the femur relative to the tibia along the flexion angle. (**B**) The internal and external rotation change of the tibia relative to the femur along the flexion angle. The means and standard deviations of anterior+/posterior–translations and internal+/external– rotation (I/E rotation) were represented by solid lines (red line: MP, green line: PS) and shaded area. The red bars along the horizontal axis indicated statistically significant differences between MP-TKA and PS-TKA. MP: Medial-pivot. PS: Posterior-stabilized. TKA: Total knee arthroplasty.

**Figure 3 bioengineering-10-00290-f003:**
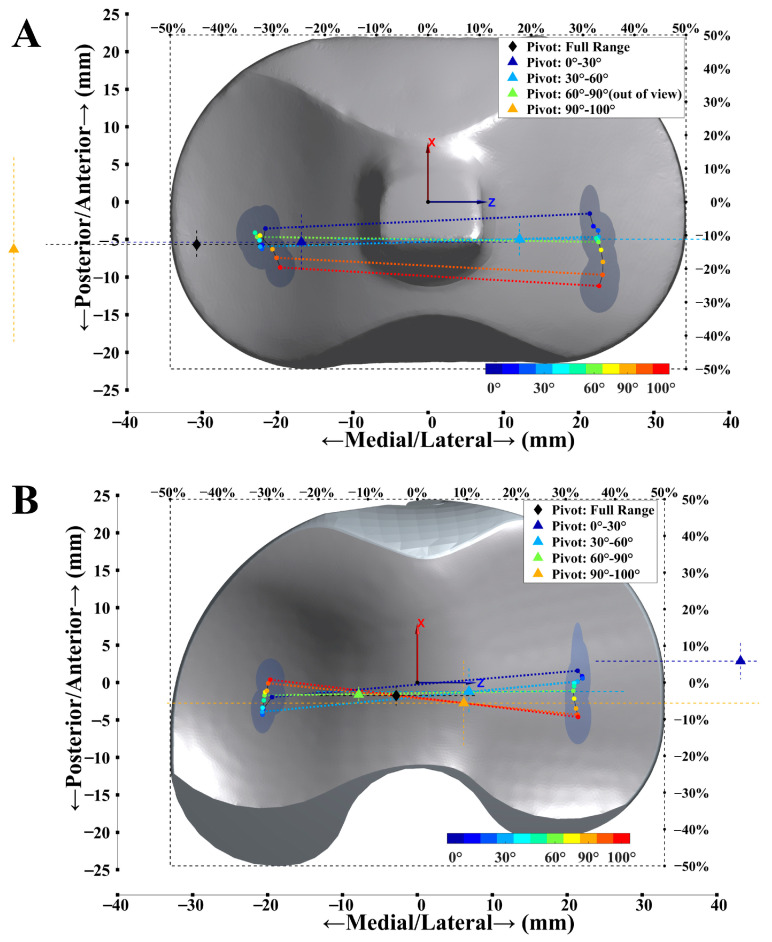
Locations of medial and lateral condylar contact points of PS-TKA (**A**) and MP-TKA (**B**) were shown on the right tibial polyethylene insertions as viewed from above during lunge. The origin was defined as the center of tibial insertions. Colored dots from blue to red represented the mean contact points in different flexion angles during the lunge. The dark blue area represented the ±1 standard deviation in medial/lateral and anterior/posterior directions. The black diamond showed the pivot point during full range lunge. The colored triangles showed the pivot point of lunge in the phase of low flexion (0°–30°), pre-middle flexion (30°–60°), post-middle flexion (60°–90°) and high flexion (90°–100°). The error bars of pivot points in the *x*-axis and the *z*-axis were illustrated with dot lines. The standard deviation of the pivot point of PS-TKA at high flexion in the *z*-axis was too large to show in the figure. Please refer to Table 3. MP: medial-pivot. PS: posterior-stabilized. TKA: total knee arthroplasty.

**Figure 4 bioengineering-10-00290-f004:**
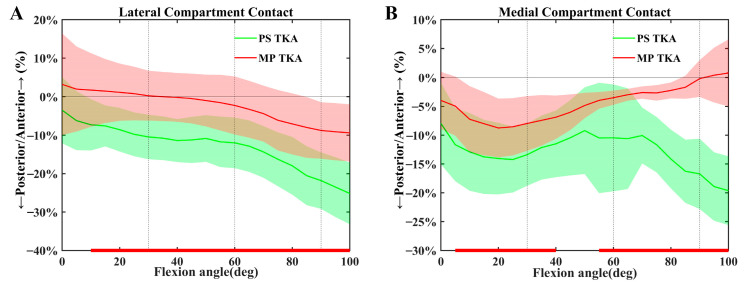
The average articular contact points in PS-TKA and MP-TKA knees were displayed through the range of flexion on lateral (**A**) and medial (**B**) compartments. Colored areas represent standard deviations. The red bars along the horizontal axis indicated statistically significant differences range (*p* < 0.0500). MP: Medial-pivot. PS: Posterior-stabilized. TKA: Total knee arthroplasty.

**Figure 5 bioengineering-10-00290-f005:**
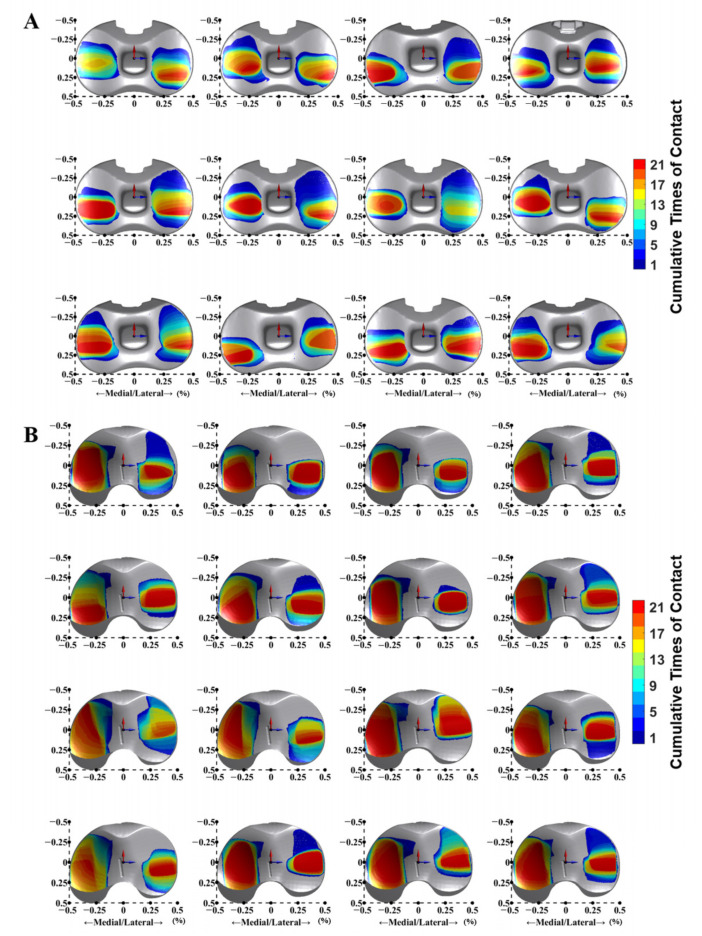
The heatmap of cumulative times of contact for ((**A**), n = 12) PS-TKA and ((**B**), n = 16) MP-TKA during lunge were presented on the right standard size of tibial insertions. The range of lunge was divided into 21 instantaneous moments with 5° flexion intervals from 0° to 100°. The dark blue area on tibial insertion showed a few contact times, while red highlighted areas with high contact times. The length and width of insertion were normalized to −0.5–0.5 (−50%–50%). MP: Medial-pivot. PS: Posterior-stabilized. TKA: Total knee arthroplasty.

**Table 1 bioengineering-10-00290-t001:** Patient demographics data.

Variables	MP-TKA	PS-TKA	*p*-Value
Age (year)	70.4 ± 5.5	67.2 ± 5.3	0.1894
Height (cm)	160.9 ± 7.1	159.6 ± 7.5	0.4037
Weight (kg)	67.5 ± 10.2	71.0 ± 13.3	0.7510
BMI (kg/m^2^)	26.9 ± 2.4	27.7 ± 3.9	0.5625
Follow-up Time (month)	11.5 ± 2.9	26.0 ± 3.9	0.0001

MP: Medial-pivot. PS: Posterior-stabilized. TKA: Total knee arthroplasty.

**Table 2 bioengineering-10-00290-t002:** Patient clinical outcome result.

Variables	MP-TKA	PS-TKA	*p*-Value
Preoperative HKA	174.3° ± 1.6°	173.5° ± 2.2°	0.2835
Postoperative HKA	178.4° ± 1.1°	177.9° ± 1.7°	0.2841
Postoperative passive range of flexion	110.9° ± 11.4°	107.1° ± 10.0°	0.5675
Knee Score of KSS	73.6 ± 10.6	77.0 ± 9.4	0.2842
Functional Score of KSS	87.5 ± 13.6	79.2 ± 15.6	0.2283
Forgotten Joint Score	73.9 ± 21.3	57.2 ± 24.0	0.1124
EQ-5D Score	1.2 ± 0.2	1.2 ± 0.3	0.9042
Patient Satisfaction Level Score	1.2 ± 0.2	1.3 ± 0.6	0.5813

Hip-knee-angle (HKA). Knee society score (KSS). MP: Medial-pivot. PS: Posterior-stabilized. TKA: Total knee arthroplasty.

**Table 3 bioengineering-10-00290-t003:** Average and standard deviation of medial and lateral contact locations of MP and PS TKA at 0°, 30°, 60°, 90° and 100° of lunge flexion along the anterior/posterior and lateral/medial direction were shown.

		MP					PS				
	Flexion Angle	0°	30°	60°	90°	100°	0°	30°	60°	90°	100°
Medial contact position	A/P (%)	−4.0 (4.9)	−8.0 (4.8)	−3.5 (1.2)	−0.2 (3.2)	0.8 (5.8)	−8.0 (7.1)	−13.4 (5.4)	−10.5 (9.3)	−16.7 (6.1)	−19.6 (6.0)
	L/M (%)	−29.4 (2.8)	−31.4 (1.9)	−30.9 (1.3)	−30.1 (1.5)	−29.7 (2.9)	−31.5 (2.9)	−32.5 (2.8)	−33.1 (3.2)	−29.4 (2.3)	−28.7 (2.9)
Lateral contact position	A/P (%)	3.2 (13.2)	0.3 (6.5)	−2.3 (7.6)	−8.8 (7.4)	−9.4 (7.4)	−3.5 (8.6)	−10.5 (5.8)	−12.0 (6.6)	−21.8 (7.3)	−25.2 (8.0)
	L/M (%)	32.4 (1.4)	32.4 (1.6)	31.5 (1.7)	32.4 (1.8)	32.5 (2.7)	31.3 (2.0)	33.1 (3.3)	33.0 (2.9)	33.8 (3.3)	33.2 (3.8)
	Flexion range	0°–30°	30°–60°	60°–90°	90°–100°	0°–100°	0°–30°	30°–60°	60°–90°	90°–100°	0°–100°
Medial contact range	A/P (mm)	4.1 (2.2)	2.5 (2.2)	1.9 (1.7) ^a^	1.6 (0.8)	6.8 (3.1)	3.1 (1.4)	3.2 (2.1)	4.5 (1.8) ^a^	1.3 (0.7)	6.6 (2.3)
	A/P (%)	8.4 (4.4)	5.0 (4.5)	3.8 (3.5) ^b^	3.3 (1.6)	13.8 (6.4)	7.1 (3.1)	7.2 (4.8)	10.1 (4.1) ^b^	3.0 (1.5)	14.9 (5.2)
Lateral contact range	A/P (mm)	4.8 (2.6)	2.1 (1.4)	3.5 (2.2)	1.0 (0.5) ^c^	9.3 (4.4)	4.1 (2.3)	2.8 (1.3)	4.6 (1.8)	1.5 (0.6) ^c^	10.5 (4.1)
	A/P (%)	9.8 (5.4)	4.2 (2.9)	7.2 (4.5)	2.0 (1.1) ^d^	19.0 (9.1)	9.2 (5.2)	6.3 (2.8)	10.4 (4.0)	3.3 (1.5) ^d^	23.7 (9.3)
Pivot point location	Flexion range	0°–30°	30°–60°	60°–90°	90°–100°	0°–100°	0°–30°	30°–60°	60°–90°	90°–100°	0°–100°
	A/P (%)	5.8 (5.0)	−2.5 (6.3)	−3.3 (16.5)	−5.6 (11.5)	−3.6 (2.7)	−12.0 (5.4)	−11.2 (4.8)	−8.1 (18.4)	−14.2 (27.6)	−12.7 (4.2)
	L/M (%)	65.4 (29.3)	10.4 (31.3)	−11.8 (1.6)	9.4 (60.0)	−4.3 (15.9)	−24.5 (61.5)	17.7 (36.0)	−112.1 (324.4)	−80.3 (228.5)	−44.8 (29.3)

Pivot point locations with respect to the insert center at different lunge flexion phases were presented. The value in ‘( )’ indicated one standard deviation value. ^a,b,c,d^ indicate a significant difference between the corresponding variables (*p* < 0.0010).

**Table 4 bioengineering-10-00290-t004:** The contact area and the 21 times contact area of PS and MP TKA patients were shown.

		Contact Area (mm^2^/%)		*p*-Value	Area with 21 Times Contact (mm^2^/%)		*p*-Value
MP	Medial	725.4 ± 54.2	(71.5 ± 5.3%)	0.0197	234.0 ± 89.5	(23.1 ± 8.8%)	<0.0001
	Lateral	530.5 ± 91.4	(46.1 ± 7.9%)		128.2 ± 44.0	(12.6 ± 4.3%)	
PS	Medial	446.7 ± 58.3	(48.7 ± 6.4%)	0.0639	96.1 ± 47.3	(10.4 ± 5.1%)	0.0136
	Lateral	519.4 ± 102.8	(56.7 ± 11.2%)		31.6 ± 21.3	(3.4 ± 2%)	

The percentage of contact area was calculated with respect to the normalized medial and lateral polyethylene insert surface area of PS and MP.

## Data Availability

Not applicable.

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
