# Peer review of "Larger Medial Contact Area and More Anterior Contact Position in Medial-Pivot than Posterior-Stabilized Total Knee Arthroplasty during In-Vivo Lunge Activity"

_bioengineering, 2023, doi:10.3390/bioengineering10030290_

Round 1

Reviewer 1 Report

The Authors aimed to investigate the in-vivo kinematics, articular contact positions, pivot point location and area of the medial and lateral compartments in MP-TKA and PS-TKA.

The topic is extremely interesting and the study is well designed.

The paper is well organized and written.

Exclusion criteria: how much values degrees?

How was decided whether to implant a MS or a PS TKA? This must be detailed.

I would split table 1 in two: baseline (methods) and outcome (results) parameters.

Conclusions are not completely supported by results ("PS insert might had a higher risk of delamination type wear"). oPlease modify. This point shold be discussed further in discussion, with relevant references. Also, a table reporting clinical outcomes of MS and PS TKA might be an added value.

Reviewer 2 Report

I command the authors for their research entitled "Larger medial contact area and more anterior contact position in medial-pivot than posterior-stabilized total knee arthroplasty during in vivo lunge activity". The manuscript is very interesting, the methods are well described, the conclusions are based on the results and the references are contemporary. I would suggest to add a picture or drawing of the movement analysed ("lunge activity") since it is crucial for the understanding of the text.  

Minor modifications:

Table 1, row 3: MBI (kg/m2) 26.9±2.4 27.7±3.9 0.56 - please check wording (BMI).

Please mention all Figures as they appear in the body of the manuscript.

Reviewer 3 Report

The authors evaluated kinematics and articular contact status between medial-pivot total knee arthroplasty (MP-TKA) and posterior stabilized TKA (PS-TKA) from 16 and 12 patients, respectively. The manuscript has some interesting findings and the hypotheses are being discussed quite comprehensively. I don't have any major issues but some comments to raise:

What does postoperative time in Table 1 mean? As there is a significant difference between MP-TKA and PS-TKA, could this affect the results? 

Introduction: Could some of the (recent) reviews on this topic be utilized in Introduction?

Methods: page 4, row 142: please rephrase.

Results: 3.1. Even though the majority of the results were insignificant, would it be informative to see p-values?

Figures 2 and 3 have not been referenced anywhere? 

Discussion page 11, rows 305-306: please rephrase

Rows 370-71: please rephrase

Row 376: follows up - please rephrase

Rows 377-378: please rephrase

Row 386: please rephrase

Round 2

Reviewer 1 Report

The paper was significantly ameliorated.

However, I still have a few concerns.

The decision on whether to implant a MS or PS TKA was based on surgeon's preferences. Nonetheless, it should be detailed some parameters considered when taking this decision.

Author Response

Reply:

Thank you for your professional comments. The clinical indications for TKA surgery were the same in both groups after our inclusion and exclusion criteria screening process. Our surgeon randomly selected the two prostheses for these patients who met our criteria. We describe our inclusion criteria in more detail at line 91-101. This manuscript compared the in vivo functional performance of the two implants in detail, which improves surgeons' understanding of these two implants. Perhaps the results of this manuscript may influence the decision of doctors in choosing the appropriate implant for patients in the future.

Line 91-101: " A senior doctor randomly determined the implant type for those patients with the similar clinical indications. Consistent inclusion and exclusion criteria were applied in the two groups. The inclusion criteria were as follows: (1) 18 to 85 years; (2) diagnosed with end-stage osteoarthritis without neuromuscular disease (Kellgren–Lawrence grade 3 and grade 4); (3) agreeing to participate in this research and signing an informed consent form before the experiment. The exclusion criteria: (1) diagnosed with valgus deformation or varus deformation over 10 degrees before TKA surgery; (2) any postoperative complications such as unbearable pain or instability, etc. before the experiment. (3) Body Mass Index > 40 kg/m2, (4) rheumatoid arthritis. (5) chronic inflammatory joint diseases, (6) patients with a pre-pathological abnormal gait. (7) pregnancy or breastfeeding.